# Antivirulence Agent as an Adjuvant of β-Lactam Antibiotics in Treating Staphylococcal Infections

**DOI:** 10.3390/antibiotics11060819

**Published:** 2022-06-17

**Authors:** Peng Gao, Yuanxin Wei, Sherlock Shing Chiu Tai, Pradeep Halebeedu Prakash, Ho Ting Venice Iu, Yongli Li, Hin Cheung Bill Yam, Jonathan Hon Kwan Chen, Pak Leung Ho, Julian Davies, Richard Yi Tsun Kao

**Affiliations:** 1Department of Microbiology, Li Ka Shing Faculty of Medicine, The University of Hong Kong, Hong Kong, China; nitawei@connect.hku.hk (Y.W.); ssct@hku.hk (S.S.C.T.); pradeep8@hku.hk (P.H.P.); veniceiu@connect.hku.hk (H.T.V.I.); liyongli1987@gmail.com (Y.L.); billyam@connect.hku.hk (H.C.B.Y.); jonchk@hku.hk (J.H.K.C.); plho@hku.hk (P.L.H.); 2Department of Microbiology, Queen Mary Hospital, Hong Kong, China; 3State Key Laboratory of Emerging Infectious Diseases and the Research Centre of Infection and Immunology, Li Ka Shing Faculty of Medicine, The University of Hong Kong, Hong Kong, China; 4Carol Yu Centre for Infection, The University of Hong Kong, Hong Kong, China; 5Department of Microbiology and Immunology, The University of British Columbia, Vancouver, BC V6T 1Z3, Canada; jed@mail.ubc.ca

**Keywords:** MRSA, antibiotics, combination, antivirulence, subinhibitory concentration, adjuvant

## Abstract

*Staphylococcus aureus* can cause a plethora of life-threatening infections. Antibiotics have been extensively used to treat *S. aureus* infections. However, when antibiotics are used at sub-inhibitory concentrations, especially for β-lactam antibiotics, they may enhance staphylococcal pathogenicity and exacerbate the infection. The combination of antivirulence agents and antibiotics may be a novel approach to controlling antibiotic-induced *S. aureus* pathogenicity. We have illustrated that under in vitro conditions, antivirulence agent M21, when administered concurrently with ampicillin, suppressed the expression and production of virulence factors induced by ampicillin. In a mouse peritonitis model, M21 reduced bacterial load irrespective of administration of ampicillin. In a bacteremia model, combinatorial treatment consisting of ampicillin or ceftazidime and M21 increased the survival rate of mice and reduced cytokine abundance, suggesting the suppression of antibiotic-induced virulence by M21. Different from traditional antibiotic adjuvants, an antivirulence agent may not synergistically inhibit bacterial growth in vitro, but effectively benefit the host in vivo. Collectively, our findings from this study demonstrated the benefits of antivirulence–antibiotic combinatorial treatment against *S. aureus* infections and provide a new perspective on the development of antibiotic adjuvants.

## 1. Introduction

The predisposition of *S. aureus* to cause infections requires the deployment of numerous virulence factors. Using these factors, *S. aureus* colonizes, disseminates, and adapts to various environments in the host and causes infections. Once they establish the infection they subvert host functions, invade tissues, and overcome host defenses [1]. Antibiotics are currently used to control the spread of *S. aureus* infections. They subside bacterial infections either by killing, decelerating, or suspending their growth. During this process, antibiotics raise pressure for the selection of resistant or tolerant strains [2]. One of the best examples is the emergence of resistance to methicillin in *S. aureus*. The resistant strains were first reported within two years after being marketed [3]. This robust emergence of resistance in pathogens like *S. aureus* has made antibiotics ineffective [4]. In addition to discovering novel antibiotics, finding antibiotic adjuvants is a promising approach to treating infections caused by resistant bacteria. Using these novel adjuvants, we can tackle the problems arising from safe and old antibiotics.

In addition to antibiotic resistance, sub-inhibitory concentrations of antibiotics may exacerbate infection by inducing staphylococcal pathogenicity [5]. For example, in previous research, we found that sub-inhibitory doses of ampicillin increased the adhesion and invasion of *S. aureus* on epithelial cells [6]. Thus, the quest for an alternate and long-lasting antivirulence agent is essential and very important. Antivirulence agents operate differently from antibiotics and have distinct benefits. For instance, the development of resistant or tolerant strains may be challenging for this class of drugs, and their administration may have no effect on the typical flora of the host [7]. In our previous study, we identified an antivirulence agent, M21, which specifically inhibited the activity of a key virulence determinant Clp protease. The inactivation of Clp protease by M21 further repressed multiple virulence factors simultaneously. M21 reduced the ratio of adherence and invasion in the host and attenuated the establishment of infection in mice by *S. aureus* [8]. Thus, different from traditional antibiotic adjuvants, antivirulence agents may compensate for the shortcomings of antibiotics and suppress *S. aureus* pathogenicity effectively.

In this work, we examined the impact of antivirulence agent M21 on staphylococcal virulence factors induced by sub-inhibitory concentrations of antibiotics. Using peritonitis and bacteremia infection models, we found that the antibiotic adjuvant M21 substantially curtailed the β-lactam-induced exacerbation of *S. aureus* infections. Thus, our findings proved that the novel antibiotic adjuvant M21 can reduce virulence induced by sub-inhibitory concentrations of antibiotics in vivo and illustrated the possibility of using a combination therapy comprising antivirulence agents and antibiotics to treat staphylococcal infections.

## 2. Results

### 2.1. M21 Repressed the Virulence Gene Expression Induced by Ampicillin

Using the luminescence-based disc diffusion assay, we detected red rings across the disc. This luminescence pattern confirmed virulence expression induced by ampicillin. Discs with M21 alone substantially repressed the luminescence signal and suggested the inhibition of *S. aureus* virulence. Open ring signals between ampicillin and M21 discs indicated that M21 significantly reduced ampicillin-induced virulence gene expression (Figure 1a).

Since ampicillin can promote *S. aureus* adherence and invasion [6], while M21 can inhibit these effects [8], a combination of ampicillin and M21 was attempted to test these phenomena. M21 reduced *S. aureus* adherence and invasion at varying concentrations, regardless of the presence of ampicillin (Figure 1b,c). This may be due to the combination of ampicillin and M21, which resulted in decreased virulence factors production.

Using Western blotting, we confirmed that the ampicillin at a concentration of 25 μM (9 μg/mL) stimulated the production of protein A and α-toxin in MRSA strain Mu3. The induction of virulence gene expression by ampicillin was suppressed in the presence of M21 at various dosages (Figure 2a). This suggests that the antivirulence agent may compromise the virulence enhancement effect of antibiotics. These findings indicated that the compound M21 might exert some control over the virulence-inducing effects of antibiotics in vivo.

### 2.2. M21 Suppressed the S. aureus Virulence Induced by Different β-Lactam Antibiotics

Supportive evidence from in vitro findings prompted us to study the role of combinatorial treatment under in vivo conditions. We first tested the in vivo efficacy of M21 in suppressing ampicillin-induced staphylococcal infection. In the mouse peritonitis infection model, on day 3, mice treated with M21 alone showed a significant reduction in the bacterial count (up to 1-log reduction), especially in livers (*p* = 0.0288) and spleens (*p* = 0.0147) (Figure 2b,c). Later using the same model, we tested the efficacy of ampicillin monotherapy and antivirulence combination therapy. When compared with ampicillin monotherapy, combinatorial treatment consisting of both ampicillin and M21 significantly reduced bacterial loads (up to 1-log reduction) in the spleen (*p* = 0.0006) and liver of mice (*p* = 0.0054) on day 6 (Figure 2d,e).

In the bacteremia model, mice treated with ampicillin alone had higher mortality than vehicle control. However, the mice treated with M21 or in combination with ampicillin recovered from the infection (Figure 3a). Thus, compound M21 suppressed the ampicillin-induced virulence in vivo. The exacerbated disease resulting from sub-inhibitory concentrations of ampicillin can be minimized by repressing *S. aureus* virulence, and the application of antivirulence agents during antibiotic therapy is highly effective in reducing antibiotic-induced pathogenicity.

In addition to ampicillin, another antibiotic belonging to β-lactams, ceftazidime was evaluated in combination with M21 to treat *S. aureus* infection. By using a mouse bacteremia model, we found that when antibiotics such as ceftazidime are used alone, they failed to save the infected mice (Figure 3b). The mice in the antibiotic treatment group died earlier than in the vehicle group. In contrast to this observation, the addition of M21 to the antibiotic therapies significantly increased mice survival rates. It is interesting to note that the compound M21 did not alter the MIC of ampicillin and suggested a novel adjuvant mechanism for antibiotics. Since there is no synergistic effect between M21 and selected antibiotics in vitro (Figure 3c), the antivirulence property of M21 appeared to be a decisive factor in reducing the antibiotic-induced virulence under in vivo conditions.

### 2.3. Combinatorial Treatment of Antivirulence Agent and Antibiotic Reduced Host Cytokine Expression

Aggravation of virulence and pathogenesis in *S. aureus* by different classes of antibiotics at sub-inhibitory concentrations hinted us to envisage the host-immune response during infection. By inducing the expression of *S. aureus* virulence factors, β-lactam antibiotics simultaneously enhanced the expression of cytokines, IL-6, and TNF-α genes in mouse kidneys. This drastic increase in proinflammatory cytokines might have contributed to an excessive systemic inflammatory response leading to the mortality of mice [9]. M21 treatment did not reduce cytokine expression when used alone. However, the antibiotic-induced immune response was suppressed when the mice were treated with M21 and antibiotics (Figure 4a,b). The total bacterial load in this combinatorial treatment was similar in different groups (Figure 4c). This suggests that antibiotics tend to induce MRSA virulence and trigger the unmanageable immune response which resulted in the death of mice. Thus, an antivirulence agent in combination with antibiotics may also help to address the antibiotic-induced host immune response during infection.

### 2.4. The Virulence Suppressing Effect of M21 Is Common in Different Strains

To further investigate the virulence suppressing effect of M21 in combination with antibiotics in different clinical isolates, we performed a paper disc diffusion assay. In this assay, M21 suppressed the virulence factors induced by the sub-inhibitory concentration of cefoxitin in 20 clinical isolates (Figure 5, Table 1, and Appendix A). These findings justify the need for antivirulence agents to control antibiotic-induced pathogenicity by *S. aureus* and serve as a novel therapeutic strategy.

## 3. Discussion

The objective of conventional drug development strategies is to discover the next chemical or molecular entity with a novel mode of action. For each promising novel substance, the journey from early discovery to market introduction is slow, expensive, and rife with obstacles. Moving a novel medicine from pre-clinical stages to the market typically involves a minimum of 10 to 12 years and costs over USD 2 billion [10]. Antibiotics, which are considered one of the greatest discoveries of humankind, received immense scientific and pharmaceutical attention in the 20th century. However, the emergence of antibiotic-resistant strains created irreparable damage not only to the pharmaceutical industry but also to the people suffering from life-threatening bacterial infections. To combat these drawbacks, the zeal for the discovery of novel antibiotics or adjuvants, or alternatives to antibiotics has not stopped. Especially, there has been tremendous progress in the discovery of novel adjuvants with the hope that they can “Make Antibiotics Great Again” [11]. However, studies have shown that the sub-inhibitory concentration of antibiotics such as ampicillin can induce virulence expression in *S. aureus.* Despite this drawback, there has been limited progress in discovering novel molecules that can control antibiotic-induced virulence. Thus, to address this lacuna, in the current study we focused on one such novel molecule called M21, which not only displayed antivirulence properties against MRSA but also functioned as an adjuvant for β-lactam antibiotics and suppressed the antibiotic-induced virulence.

Combination regimens for the prevention or reversal of antibiotic resistance are already a widespread method. They usually consist of two drugs, an antibiotic plus an antibiotic adjuvant [12]. A combination of an antibiotic and an antibiotic adjuvant varies from a combination of two antibiotics. In the former, the adjuvant may have little to no in vitro action against bacteria, particularly at clinically relevant levels [13]. The adjuvant’s principal function is to improve antibiotic activity. Additionally, these adjuvants may act as anti-resistant or antivirulence molecules by targeting specific pathways in the bacterial system. For example, adjuvants are reported to inhibit quorum sensing pathways in bacteria and prevented the formation of biofilm under in vitro and in vivo conditions [7,14]. Owing to these findings, the antivirulence properties of M21 were reassessed to address the bacterial virulence induced by antibiotics.

Virulence in *S. aureus* is mediated by several factors such as surface proteins, toxins, and superantigens. Among these factors, surface proteins contribute to the initial adhesion and colonization of bacteria to the host surface and promote bacterial virulence. Once they colonize, toxins attack the host immune cells and favor the survival of *S. aureus* in the host, leading to life-threatening infections [15]. Thus, combating *S. aureus* infection through virulence suppression by the non-antibiotic method is a promising alternative approach [16,17]. Thus, in the present study, the combination of antibiotics and an antivirulence agent was tested to control antibiotic-induced *S. aureus* virulence. 

β-lactams comprise a class of therapeutically significant antibiotics that are commonly employed as the first line of treatment for infectious diseases. However, more than 75% of patients with MRSA infections were shown to receive inappropriate agents for initial antimicrobial treatment [3]. Due to the differences in tissue distribution, clearance, and metabolism rates, even when suitable antibiotics and dosages are utilized, the concentrations in certain organs of the human body might be lower than the MIC for some time periods [18]. Antibiotics that are used for an extended period of time and at a low dose can develop resistance and increase pathogenicity. At non-lethal concentrations, β-lactam antibiotics induce the production of α-toxin and PVL toxin, toxic shock syndrome toxin (TSST), enterotoxins, and LukED, which should contribute to the alteration of therapy and, therefore, contribute to worse outcomes [19,20]. Induced bacterial adhesion by antibiotics commonly used in the management of *S. aureus* infections may encourage invasive intracellular strains, which may have an important effect on the persistence and recurrence of infections [3]. Numerous investigations established a relationship between methicillin resistance and poor clinical outcomes. MRSA strains were not shown to be considerably more virulent than MSSA strains [21]. If antibiotic exposure is included, a resistant strain is more likely to be exposed to an unsuitable substance, resulting in greater pathogenicity [22]. By introducing an antivirulence agent, we can control the antibiotics-induced virulence, to manage MRSA infection. A combination of β-lactam antibiotics and an antivirulence drug is preferable than using β-lactam antibiotics alone because of the risk of a worse outcome [9].

The combination of antibiotics and antivirulence agents may harness the full potential of both agents [23]. A combination of the antivirulence agent with antibiotic was evaluated with various in vitro molecular, biochemical, and cellular assays and in vivo mouse models. Using the bacteremia model, we tested the induction of virulence by sub-inhibitory concentrations of ceftazidime (β-lactam antibiotics) and confirmed the worsening of infection by MRSA in mice. Infected mice treated with these antibiotics alone not only failed to reduce the bacterial load but also heightened the host immune response by enhancing bacterial virulence. Some toxins, such as PSMs, enhance the release of cytokines CXCL8, CCL20 IL-6, and TNF-α [24]; α-toxin stimulate cytokines, such as nuclear factor-kB (NF-kB), IL-6, but not TNF-α [25]; protein A will induce IL-8, TNF-α, and MIP-1α. In the present study, we demonstrated that the antivirulence molecule M21 could profoundly reduce α-toxin and protein A expression induced by subinhibitory concentrations of antibiotics [26]. Additionally, the antivirulence agent M21, in combination with antibiotics, significantly improved the survival of mice in the bacteremia model and reduced the *S. aureus* virulence expression in vivo. Additionally, by repressing the antibiotic-induced bacterial virulence, this novel combinatorial therapy reduced the host immune response during the infection. Thus, findings from the present study not only epitomize the dark side of antibiotics but also provide us an opportunity to reform our understanding of concealed virulence observed in *S. aureus* strains. One of the intriguing approaches to address this challenge is the inclusion of antivirulence agents in the treatment regimen.

Currently, most of the research focused on the discovery of novel antibiotic adjuvants to sensitize resistant bacteria. After restoring the susceptibility to antibiotics, such as β-lactam antibiotics and aminoglycosides, the old drugs can be used again. Antibiotics adjuvants will show a synergistic effect with certain antibiotics by targeting the resistance genes or increasing the intake or inhibiting the efflux pumps. It is noteworthy to mention that the compound M21 did not show any synergistic effects with the tested antibiotics and reduced the antibiotic-induced virulence independently. This suggests that M21 does not interfere with antibiotics but works independently by suppressing multiple virulence factors. Hence, an antivirulence molecule such as M21 differs completely from traditional antibiotic adjuvants which are known to potentiate the activity of antibiotics. Thus, our findings using this small molecule, justify a novel perspective of developing antibiotics adjuvant and the need for antivirulence agents in suppressing antibiotic-induced virulence.

In concordance with these findings, our studies provide explicable evidence that the exacerbated infections caused by sub-inhibitory concentrations of antibiotics can be controlled by an antivirulence agent. We found that the compound M21 could overcome the virulence induction caused by ampicillin and ceftazidime (and presumably some β-lactam antibiotics as well). Thus, our studies illustrate the necessity for non-antibiotic-based antivirulence agents in suppressing *S. aureus* virulence. The application of these compounds in combination with antibiotics will possess high therapeutic values, and this could be a better choice to control *S. aureus* infection globally.

## 4. Materials and Methods

### 4.1. Bacterial Strains and Plasmids

The bacterial strains used in this study are listed in Table 2. Brain heart infusion (BHI) broth and BHI agar plates were used to grow *S. aureus*. Chloramphenicol was used at 10 µg/mL. Unless otherwise stated, all cultures were grown aerobically at 37 °C with shaking at 250 rpm and growth was monitored at 600 nm with a HITACHI U-2800 (Hitachi, Japan) spectrophotometer.

### 4.2. Minimum Inhibitory Concentration (MIC) Tests

In 96-well plates, the MIC was measured by inoculating 5 × 10^4^
*S. aureus* cells in 100 μL BHI medium with a serial dilution of antibiotics. After 18 h at 37 °C, the MIC was determined as the lowest concentration that resulted in a cell density of less than 0.01 OD at 620 nm, indicating no observable growth.

### 4.3. Disk Diffusion and Lux Assays

A single colony of bioluminescent *S. aureus* strains containing plasmid pGL*hla* (Table 2) from BHI agar was resuspended in 200 μL sterile water, diluted to 75 mL 0.7% (*w*/*v*) soft agar, and plated on BHI agar. On the overlay, antibiotic discs (diameter 6 mm; Advantec Co., Tokyo, Japan) were placed, and the plates were incubated at 37 °C. After 20 h, inhibitory zones were determined, and luminescence was detected using a PE IVIS Spectrum in vivo imaging system (PerkinElmer, Hong Kong, China) [6]. When combining antibiotics and compounds, antibiotics (10 mM, 5 μL) and M21 (50 mM, 5 μL) were placed on the plates at a distance of 1.5 cm and 2 cm, respectively. When analyzing the interaction between cefoxitin and M21, we used a concentration of 4 mM of antibiotic and M21 and the distance between the discs was 1.5 cm.

### 4.4. Real-Time PCR to Verify Expression Levels

The preparation of total RNA from *S. aureus* was performed using an RNA protection reagent according to the manufacturer’s instructions (Qiagen, Hilden, Germany) [8].

To study the expression of mouse cytokines by q-PCR, we acquired cDNA as described above. The relative quantification of IL-6 and TNF-α transcripts was determined by the ratio of expression of target transcripts relative to Hprt gene (housekeeping gene). The sequences of primers for real-time PCR experiments are provided in Table 3.

### 4.5. Adherence Assay and Invasion Assay 

As previously described [6], before inoculation, overnight bacterial cultures treated with ampicillin (0.2 g/mL) and/or compound M21 (10 μM or 50 μM) were washed three times with PBS (pH 7.4) and diluted to 10^7^ CFU/mL with MEM medium (defined as the original bacterial CFU). A549 cells were seeded at a concentration of 2 × 10^5^/mL in MEM onto a 24-well tissue culture plate (Greiner) for measuring adherence and invasion ratio. Briefly, confluent monolayers of A549 cells were cultivated overnight at 37 °C in 5% CO_2_ to form. The next morning, the medium was removed and A549 cells were washed twice with 1 mL of PBS before being infected with 1 mL of the prepared bacterial inoculum. For the invasion experiment, after infecting A549 cells at 37 °C for two hours, the well supernatants were collected for bacterial count (defined as the total bacterial CFU). All wells were then washed three times with 1 mL PBS after which A549 cells were incubated for one hour at 37 °C in MEM containing gentamicin (100 μg/mL; Sigma-Aldrich, St. Louis, MO. USA) and lysostaphin (10 μg/mL; Sigma-Aldrich, St. Louis, MO. USA). Afterward, the wells were trypsinized with 150 μL of 0.25% trypsin-EDTA for 5 min, the cells in each well were carefully collected into tubes, and 400 μL of ice-cold 0.025% Triton X-100 was added to the tubes, which were then placed on ice.

The number of bacterial CFU released from lysed epithelial cells was evaluated by plating lysates on BHI agar plates (the invaded bacterial CFU). After infecting A549 cells at 37 °C for one hour for the adherence test, the medium was withdrawn, and the cells were washed three times with 1 mL PBS. The adherent bacterial CFU was then defined as the total number of attached and invasive bacteria liberated from the lysed epithelial cells.
$$\mathrm{Relative}\;\mathrm{invasion}\;=\frac{\mathrm{Internalized}\;\mathrm{bacteria}\;\mathrm{CFU}\;\mathrm{of}\;\mathrm{sample}/\mathrm{Total}\;\mathrm{CFU}\;\mathrm{of}\;\mathrm{sample}}{\mathrm{Internalized}\;\mathrm{bacteria}\;\mathrm{CFU}\;\mathrm{of}\;\mathrm{control}/\mathrm{Total}\;\mathrm{CFU}\;\mathrm{of}\;\mathrm{control}}$$

The bacterial adhesion in each well is given as a percentage of the CFU in the inoculum that adhered to and penetrated the cells. The control wells had just medium as a pre-treatment (MEM). Using the formulae, adhesion and invasion were then standardized versus controls.
$$\begin{array}{l} \mathrm{Relative}\;\mathrm{adherence}\;\\ =\frac{\mathrm{Adhered}\;\&\;\mathrm{Internalized}\;\mathrm{bacteria}\;\mathrm{CFU}\;\mathrm{of}\;\mathrm{sample}/\mathrm{Original}\;\mathrm{CFU}\;\mathrm{of}\;\mathrm{sample}}{\mathrm{Adhered}\;\&\;\mathrm{Internalized}\;\mathrm{bacteria}\;\mathrm{CFU}\;\mathrm{of}\;\mathrm{control}/\mathrm{Original}\;\mathrm{CFU}\;\mathrm{of}\;\mathrm{control}} \end{array}$$

Each experiment was conducted three times, and all relative adhesion and invasion data were calculated and statistically evaluated using SigmaPlot software 11.0 (Sigma Plot Software, Jandel, Chicago, IL, USA) and the Student’s t-test. Based on the *p* values, statistical significance was assessed; *p* < 0.05 was deemed significant.

### 4.6. Western Blot

*S. aureus* strains were cultured in BHI broth. This allowed us to collect the supernatants at different time points. Supernatants of ampicillin or compound M21 treated *S. aureus* was isolated after 24 h, adjusted to an OD_600_ of 6.0, with BHI, and centrifuged. After boiling in Laemmli sample buffer, 5 μL of culture supernatant was loaded onto a 12% sodium dodecyl sulfate-polyacrylamide gel. Alpha-hemolysin was detected with rabbit anti-staphylococcal α-hemolysin antibody (1:20,000) (Sigma-Aldrich, St. Louis, MO. USA) and goat Horseradish Peroxidase (HRP)-conjugated anti-rabbit IgG (1:5000) (Sigma-Aldrich). Protein A was visualized with HRP-conjugated Rabbit anti-staphylococcal Spa antibody (1:20,000) (Abcam, Cambridge, MA, USA). The Western blot protocol was performed as described in the product guide for Amersham ECL Western blotting detection reagents (GE Healthcare, Buckinghamshire, UK).

### 4.7. Mouse Peritonitis Model

As previously described [27], we kept the 6- to 8-week old BALB/c female mice in biosafety level 2 animal facility. Mice were housed in microisolator cages and received food and water ad libitum. All experimental protocols (CULATR 3055-13 and 3678-15) followed the standard operating procedures of the approved biosafety level 2 animal facilities and were approved by the Animal Ethics Committee.

Mid-exponential phase of *S. aureus* culture was washed twice with sterile PBS and resuspended again in PBS to obtain 1 × 10^8^ cfu/100 μL. Mice were i.p. injected with 4 × 10^8^ suspended early stationary phase *S. aureus*. After six hours post-infection, mice were randomized into two groups (n = 12). Every day, they were treated with 100 μL PBS or 8 mg/mL ampicillin in PBS subcutaneously (s.c.) twice (12-h interval). The third control group (n = 6), without bacterial infection, was treated only with ampicillin. Animals were daily monitored for symptoms of disease (body weight drop, inactivity, ruffled fur, and labored breath) and death.

To assess the antivirulence property of the compound M21, at the beginning of the experiment, mice were randomized into two groups (n = 10) consisting of vehicle control group (receiving PBS with 5% DMSO and 2% tween 80) and treatment group (receiving M21). Each group was treated with designated concentrations of compound M21 (3.85 mg/kg/dose) or vehicle control. Prior to infection, mice were pretreated with two doses of M21 one day before infection. On day 0, infection in mice was established by i.p. injection of 4 × 10^8^ early stationary phase *S. aureus*. Two doses of compound M21 with an interval of 12 h were administrated. On day 3, 10 animals from each group were euthanized, kidneys, livers, and spleens were harvested, homogenized in PBS, and plated on BHI agar for counting the viable bacteria.

For studying the efficacy of combinatorial treatment, before the beginning of the experiment, mice were randomized into two groups (n = 12 for ampicillin group and n = 14 for combination treatment group). They were either administered with the designated concentrations of compound M21 or vehicle control. On day 0, mice were i.p. injected with 4 × 10^8^ early stationary phase *S. aureus* mu3. On day 1, two doses of M21 pretreatment were given by i.p. injection. Six hours later, mice were treated with 100 μL of 8 mg/mL ampicillin in PBS subcutaneously. The ampicillin group was treated with the ampicillin by s.c. and vehicle group received the vehicle control by i.p., whereas, the combinatorial treatment group received ampicillin by s.c. and M21 by i.p. All these treatments comprised two dosages per day (12 h interval). The animals were daily monitored for symptoms of disease (body weight drop, inactivity, ruffled fur, and labored breath) and death.

On day 6, 16 animals were euthanized, kidneys, livers, and spleens were harvested, homogenized in PBS, and plated on BHIA for determining the bacterial viable count.

### 4.8. Mouse Bacteraemia Model 

As previously described [8], *S. aureus* strain mu3 was cultured to the early exponential phase, washed twice with sterile PBS, and resuspended in PBS to attain a cell density of 1 × 10^8^ CFU/100 μL. This bacterial suspension was used to establish the lethal mice infection model. The female BALB/c mice, 6–8 weeks, were infected through tail vein (i.v.) with *S. aureus* and randomized into 4 groups consisting of 5–12 mice per group. One hour post-infection, mice were treated with designated concentrations of antibiotics (s.c.) or compound M21 (3.85 mg/kg/dose) (i.p.) or combination of antibiotics and M21 or vehicle, serving as control. Antibiotics such as ampicillin (40 mg/kg/dose), and ceftazidime (16.5 mg/kg/dose) were used for the treatment twice per day. To study the combinatorial effect of M21 and antibiotics, we used four groups of mice for vehicle, M21, antibiotic, and combined treatments, respectively. M21 or injection buffer treatments were performed twice per day at 12 h intervals. The survival was monitored according to the body condition scoring system.

For q-PCR studies, samples were obtained from mice that had undergone survival experiments. On day 2, animals from each group were euthanized and kidneys were collected. The kidney from each mouse was divided into two halves; one half of the kidney was stored in liquid nitrogen for RNA extraction, and the other half was homogenized in PBS and plated on BHIA for determining the bacterial viable count.

### 4.9. Statistics

Statistical analysis was performed using Graph Pad Prism version 7.0. All error bars depict the standard error of mean (SEM). Horizontal lines depict the mean. All replicates are biological (from different samples). Non-parametric tests (Mann–Whitney test) were used for significant analysis. For bacteraemia model, data were analyzed by survival analysis, and Log-rank (Mantel–Cox) test was used for significance analysis.

## Figures and Tables

**Figure 1 antibiotics-11-00819-f001:**
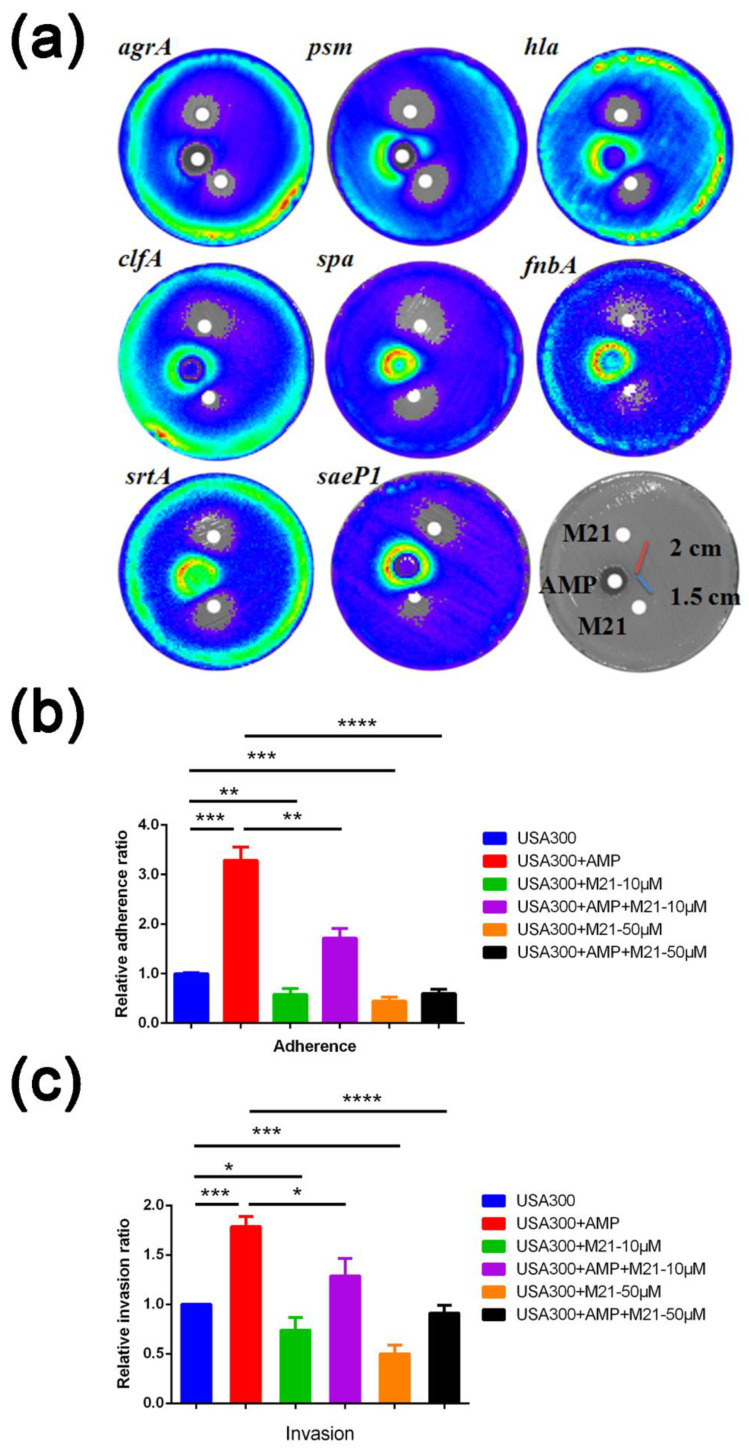
M21 reduced the *S. aureus* virulence induced by antibiotics. (**a**) The interaction of compound M21 and ampicillin on eight different promoters. Paper discs with 5 μL of 10 mM ampicillin were placed in the middle; the other two are paper discs with 5 μL of compound M21 (50 mM). The distance between the paper discs is shown in the figure. (**b**,**c**) Effect of compound M21 on the ampicillin-enhanced adherence (**b**) and invasion (**c**) of USA300 in A549 cells. Relative invasion and relative adherence assays were performed in triplicate and experiments were repeated twice. Using non-parametric tests, the treated groups were compared with the control group. Data represent mean values ± SEM (* *p* < 0.05; ** *p* < 0.01; *** *p* < 0.001; **** *p* < 0.0001).

**Figure 2 antibiotics-11-00819-f002:**
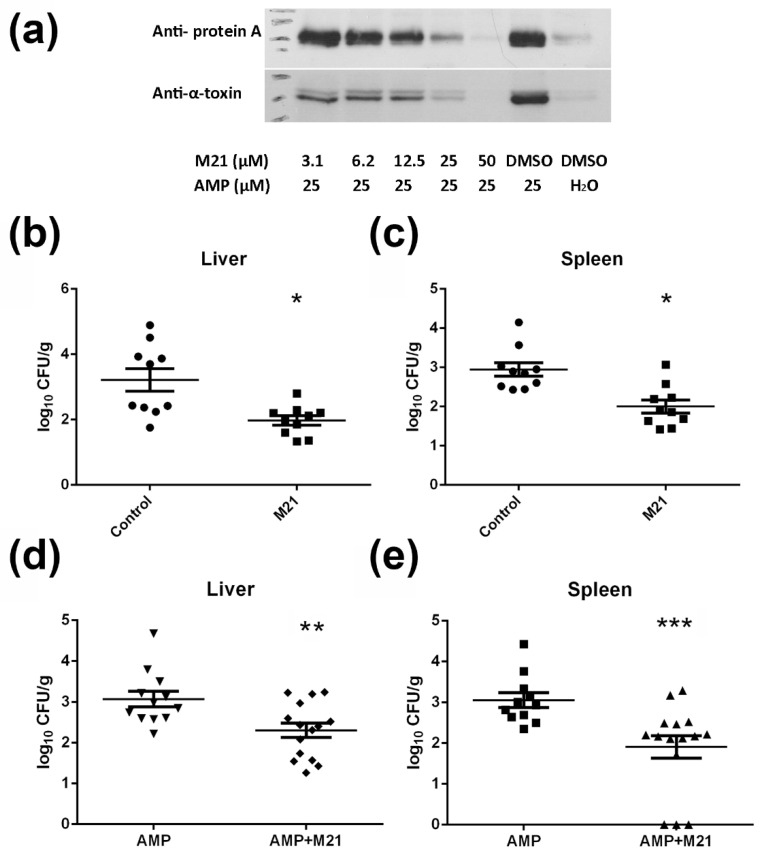
Compound M21 suppress the production of virulence factors induced by ampicillin and reduces *S. aureus* pathogenicity in vivo. (**a**) Ampicillin-induced production of protein A and α-toxin was suppressed by different concentrations of compound M21. (**b**,**c**) M21 treatment reduced bacterial loads in mice, liver (**b**), and spleen (**c**). (**d**,**e**) M21 in combination with ampicillin treatment, reduced bacterial loads in mice, liver (**d**), and spleen (**e**). Non-parametric tests (Mann–Whitney test) for treated groups, comparing with the vehicle group. Data represent mean values ± SEM (* *p* < 0.05; ** *p* < 0.01; *** *p*<0.001).

**Figure 3 antibiotics-11-00819-f003:**
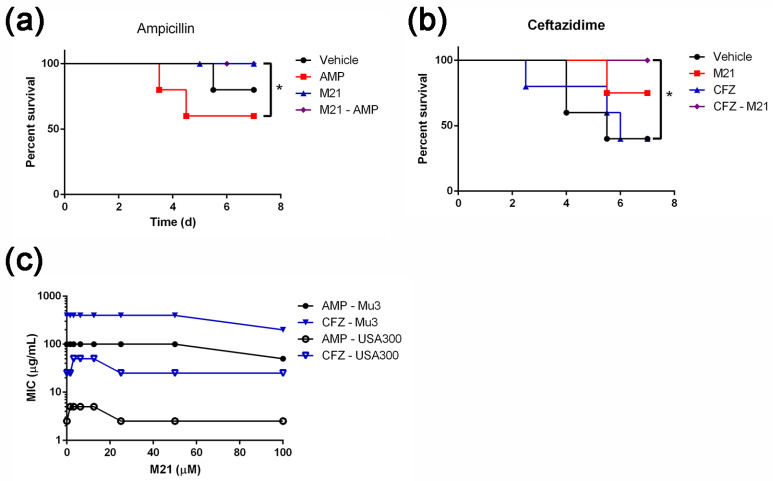
Combination of M21 with ampicillin or ceftazidime increased mice survival. (**a**) Ampicillin combined with M21 improved the survival of mice suggesting diminished antibiotic-induced virulence in *S. aureus* (MIC of M21 to Mu3 is higher than 500 µM). (**b**) Ceftazidime combined with M21 saved mice from antibiotic-induced *S. aureus* virulence. (**c**) No synergistic effect was detected among M21 and ampicillin, or ceftazidime against USA300 and Mu3 strains. Log-rank (Mantel–Cox) test was used for survival analysis and the treated groups were compared with the control group. Data represent mean values ± SEM (* *p* < 0.05).

**Figure 4 antibiotics-11-00819-f004:**
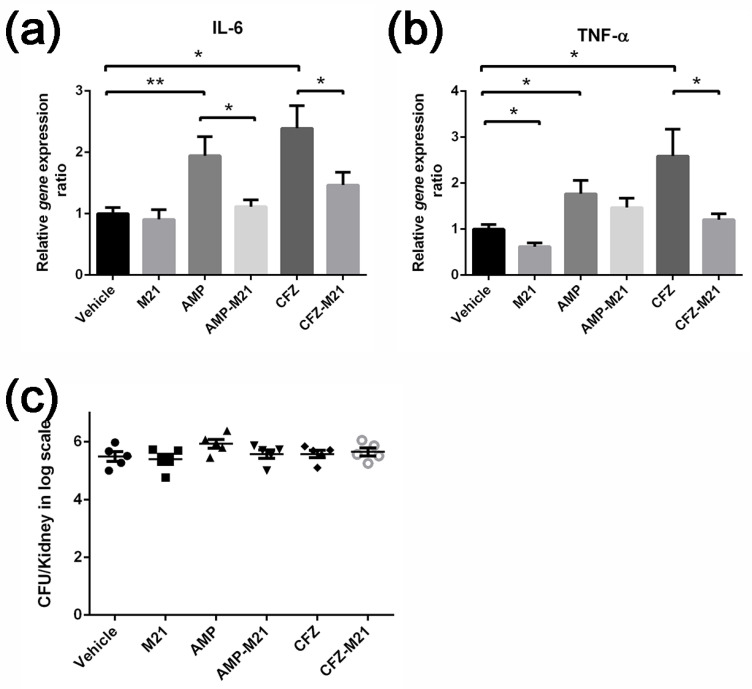
Cytokine gene expression in response to antibiotic-induced bacterial virulence. (**a**,**b**) q-PCR analysis of cytokine expression in mouse kidneys after bacterial infection and antibiotics treatment. Gene expression levels were relative to the hprt gene. (**a**) IL-6; (**b**) TNF-α; (**c**) On day 2, ceftazidime and ampicillin in combination with M21 did not increase bacterial load in bacteremia model. Non-parametric tests were performed, and the treated groups were compared with the vehicle group. Data represent mean values ± SEM (* *p* < 0.05; ** *p* < 0.01).

**Figure 5 antibiotics-11-00819-f005:**
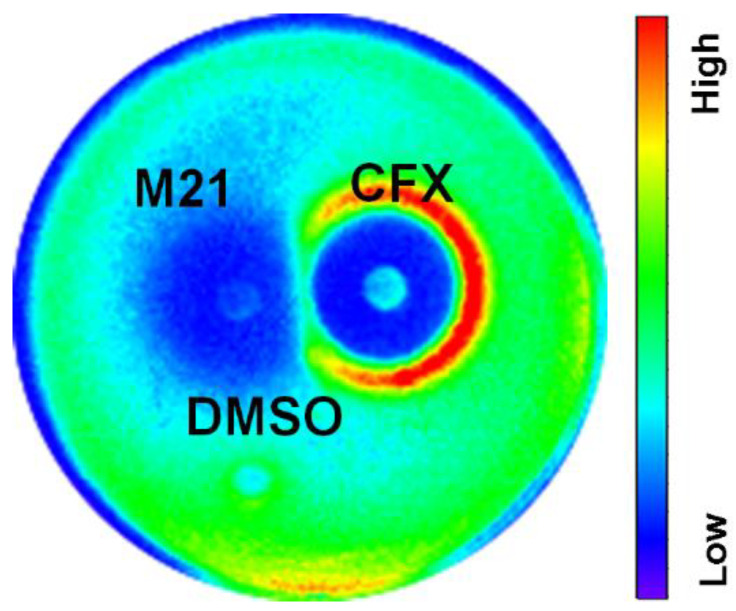
Paper disc assay showing the interaction of M21 with cefoxitin in isolate 14. Cefoxitin (4 mM, 5 µL) induced *hla* expression against different clinical isolates. M21 (4 mM, 5 µL) repressed *hla* expression even in the induced state.

**Table 1 antibiotics-11-00819-t001:** Interaction between M21 and cefoxitin in different isolates *.

Isolates	Cefoxitin Induced Virulence	M21 Antivirulence Effect	Interaction
Isolate 14	5	−6	0
Isolate 86	1	−5	0
Isolate 22	5	−6	0
Isolate 24	9	−6	0
Isolate 25	4	−3	1
Isolate 34	2	−8	0
Isolate 42	3	−7	0
Isolate 43	5	−8	0
Isolate 44	2	−3	1
Isolate 45	4	−5	0
Isolate 46	5	−3	0
Isolate 63	2	−2	1
Isolate 64	2	−2	1
Isolate 65	6	−4	1
Isolate 72	3	−2	1
Isolate 73	6	−5	1
Isolate 76	9	−5	−1
Isolate 83	1	−1	−2
Isolate 84	7	−3	0
Isolate 85	1	−8	0

*: 1 to 9: induction; −9 to −1: repression.

**Table 2 antibiotics-11-00819-t002:** Strains and plasmids used in this study.

Strain	Phenotype	Source
Lab strains		
USA300 FPR 3757	CA-MRSA, Agr+	ATCC ABB1776
Mu3	MRSA, Agr+	ATCC700698
Clinical isolates		
Isolate 14	Clinical isolate from patient blood, MRSA	This study
Isolate 22	Clinical isolate from patient blood, MSSA	This study
Isolate 24	Clinical isolate from patient blood, MRSA	This study
Isolate 25	Clinical isolate from patient blood, MRSA	This study
Isolate 34	Clinical isolate from patient blood, MRSA	This study
Isolate 42	Clinical isolate from patient blood, MRSA	This study
Isolate 43	Clinical isolate from patient blood, MSSA	This study
Isolate 44	Clinical isolate from patient blood, MSSA	This study
Isolate 45	Clinical isolate from patient blood, MRSA	This study
Isolate 46	Clinical isolate from patient blood, MRSA	This study
Isolate 63	Clinical isolate from patient blood, MSSA	This study
Isolate 64	Clinical isolate from patient blood, MRSA	This study
Isolate 65	Clinical isolate from patient blood, MSSA	This study
Isolate 72	Clinical isolate from patient blood, MRSA	This study
Isolate 73	Clinical isolate from patient blood, MRSA	This study
Isolate 83	Clinical isolate from patient blood, MRSA	This study
Isolate 84	Clinical isolate from patient blood, MSSA	This study
Isolate 85	Clinical isolate from patient blood, MRSA	This study
Isolate 86	Clinical isolate from patient blood, MRSA	This study
Plasmid		
pGL	gfp-luxABCDE dual reporter plasmid	Lab stock
pGL*hla*	gfp-luxABCDE dual reporter driven by *hla* promoter	Lab stock

**Table 3 antibiotics-11-00819-t003:** Primers used in this study.

Gene	Primer for Real-Time PCR
rt-*hprt*-f	CTGGTGAAAAGGACCTCTCG
rt-*hprt*-r	TGAAGTACTCATTATAGTCAAGGGCA
rt-*Tnf*-α-f	CTCCAGGCGGTGCCTATGT
rt-*Tnf*-α-r	GAAGAGCGTGGTGGCCC
rt-*Il*-6-f	CCAGAAACCGCTATGAAGTTCC
rt-*Il*-6-r	TCACCAGCATCAGTCCCAAG

## Data Availability

Not applicable.

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
