# Peer review of "Antivirulence Agent as an Adjuvant of β-Lactam Antibiotics in Treating Staphylococcal Infections"

_antibiotics, 2022, doi:10.3390/antibiotics11060819_

Round 1
Reviewer 1 Report
1.compound M21 means??
2.Adherence assay and invasion assay why choose A549 lung cancer cells>?
3.bacteria name should be italics
4.in vitro should be italics
5.how to set the antibiotic and compound concentrations for treatment
6.why particularly choose ambicilin.choloromphenical and ceftazidime antibiotics??
7.author should be mention the isolation (Isolate 14 to 86)source of MRSA in table 1
8.author should be add the mice histological,QPCR and RNA band images
9.add westernblot results images
10.most of the references very old,update the recent references
11.discussion very poor,improve the discussion part
12.author should be add the Adherence assay and invasion assay A549 lung cancer cells images
Author Response
1.compound M21 means??
Reply: M21 is an anti-virulence compound which we have identified previously (Gao, P.; Ho, P.L.; Yan, B.; Sze, K.H.; Davies, J.; Kao, R.Y.T. Suppression of Staphylococcus aureus virulence by a small-molecule compound. Proc Natl Acad Sci U S A 2018, 115, 8003-8008). We have added the information of M21 in the introduction part of the manuscript.
2.Adherence assay and invasion assay why choose A549 lung cancer cells>?
Reply: Work by others (Liang, X.; Ji, Y. Comparative analysis of staphylococcal adhesion and internalization by epithelial cells. Methods Mol Biol 2007, 391, 145-151) have shown that Staphylococcus aureus can invade different types of nonphagocytic cells, which in turn may contribute to the evasion of the killing by certain antibiotics. Based on the reference, we employed the same epithelial cell for this assay.
3.bacteria name should be italics
Reply: Yes, we have revised them.
4.in vitro should be italics
Reply: Yes, we have revised them.
5.how to set the antibiotic and compound concentrations for treatment
Reply: For antibiotics, we used the same doses for human (to make sure they have the same dose/kg).
For compound M21, we used the optimized concentration detected in the serum which would not affect bacteria growth in vitro as previously described (Gao, P.; Ho, P.L.; Yan, B.; Sze, K.H.; Davies, J.; Kao, R.Y.T. Suppression of Staphylococcus aureus virulence by a small-molecule compound. Proc Natl Acad Sci U S A 2018, 115, 8003-8008).
6.why particularly choose ambicilin.choloromphenical and ceftazidime antibiotics??
Reply: We used ampicillin and ceftazidime because they belong to beta-lactam antibiotics, and they have been reported to induce bacterial virulence in vitro as mentioned in our manuscript.
For chloramphenicol, we use it as selection antibiotics to maintain plasmid for paper disc assay, but not in animal experiments.
7.author should be mention the isolation (Isolate 14 to 86)source of MRSA in table 1
Reply: We added the information to table 1.
8.author should be add the mice histological,QPCR and RNA band images
Reply: We did not perform histology experiment for this experiment; hence this term will be deleted.
Figure 4a and b show q-PCR results using RNA extracted from the kidney of mice.
For the RNA band, we apologize but we did not run a gel for the RNA experiment; instead, we analyzed the concentration and purity using a NanoDrop spectrophotometer.
9.add westernblot results images
Reply: The corresponding Western blot results were depicted in figure 2a.
10.most of the references very old,update the recent references
Reply: We have updated the references.
11.discussion very poor,improve the discussion part
Reply: We have revised the discussion part.
12.author should be add the Adherence assay and invasion assay A549 lung cancer cells images
Reply: In this assay, we wanted to investigate if compound M21 could repress the induced S. aureus adherence and invasion by antibiotics quantitively. Following the reference (Liang, X.; Ji, Y. Comparative analysis of staphylococcal adhesion and internalization by epithelial cells. Methods Mol Biol 2007, 391, 145-151), we used the adherence and invasion ratios as reliable evaluation of the compound’s effectiveness.
Reviewer 2 Report
Interesting study--some rewording (line 46-52 and line58-65) for clarity and perhaps some streamlining of paper disk assay figures.

Author Response
- rewording line 46-51 and line 58-65.
Reply: We have reworded this part in the manuscript.
- change figure 5 to using represent strain and a table.
Reply: We have revised figure 5 according to your suggestions and added this full graph as a supplementary image.
Reviewer 3 Report
The study has confirmed the role of M21 as an anti-virulence agent both in vitro and in vivo. The authors have not studied the underlying mechanism affecting virulence. For example, in previous studies, it has been proposed that Quorum sensing (QS) is involved in virulence and resistance to antibiotics. In this study, the authors have not explored such mechanisms to answer the changes occurring at the molecular level. This information would have helped the reader assess the exact role of M21.
Minor Comments:
The title of the manuscript is not appropriate.
The introduction section did not include any information about M21.
The paper disc assay should have been described further.
The authors could have performed an FIC (fractional inhibitory concentration) experiment to determine if the combination of antibiotics with M21 had a synergistic or antagonistic effect.
The authors should explain how the antibiotic-induced immune response was suppressed.
The dose of 1x108cfu/100ul was too high. Why was such a high dose selected for inducing infection?
Why is the vehicle group composed of 5% DMSO and 2% Tween in PBS (Section 4.7)?
Why were the mice sacrificed on days 3 and 6 for bacterial counts while day 2 for histopathology?
While the authors mentioned that they collected histopathology samples in the methods section (Section 4.8), they did not include these results in the manuscript.
The language of the manuscript needs improvement.
Author Response
Minor Comments:
- The title of the manuscript is not appropriate.
Reply: We have changed the title of the manuscript to “Anti-virulence agent as an adjuvant of β-lactam antibiotics in treating staphylococcal infections”
- The introduction section did not include any information about M21.
Reply: M21 is an anti-virulence compound which we have identified previously (Gao, P.; Ho, P.L.; Yan, B.; Sze, K.H.; Davies, J.; Kao, R.Y.T. Suppression of Staphylococcus aureus virulence by a small-molecule compound. Proc Natl Acad Sci U S A 2018, 115, 8003-8008). We have added the information of M21 in the introduction part of the manuscript.
- The paper disc assay should have been described further.
Reply: details are added to the method part 4.3
- The authors could have performed an FIC (fractional inhibitory concentration) experiment to determine if the combination of antibiotics with M21 had a synergistic or antagonistic effect.
Reply: Yes, what we have done is FIC experiment, and we just revised our presentation format to make the result clear. Now we have presented 4 sets of checkerboard assay data in one single graph (Figure 3C). FIC index cannot represent that there are no difference in MIC of antibiotics combined with different concentration of M21.
Meanwhile, M21 has no MIC in this assay, hence we cannot calculate the FIC based on the traditional equation.
- The authors should explain how the antibiotic-induced immune response was suppressed.
Reply: We have added this in the discussion part.
- The dose of 1x108cfu/100ul was too high. Why was such a high dose selected for inducing infection?
Reply: Actually, mice without immune repressor requires a lethal dose of around 1 x 107 CFU of virulent S. aureus strain. Low virulent strain may need 1 x 108 CFU or even higher inoculum. When immune repressor, such as mucin is employed, then the lethal dose can be decreased for 3 logs. However, since the immune response is corelated to bacterial virulence, we can not use immune repressor for our purpose. Due to this reason we used high dose of bacteria for this type of animal infection model.
- Why is the vehicle group composed of 5% DMSO and 2% Tween in PBS (Section 4.7)?
Reply: We needed to use this formula (5% DMSO and 2% Tween in PBS) to dissolve compound M21, and the same composition was used for all groups including the vehicle group.
- Why were the mice sacrificed on days 3 and 6 for bacterial counts while day 2 for histopathology?
Reply: For the peritonitis model, mice are typically sacrificed on day 3. After day 3, the immune system of mice will rapidly eliminate the bacteria. However, antibiotics may stimulate bacterial virulence and prolong bacterial infection in mice until day 6. Thus, to obtain a significant difference between the ampicillin and combination therapy groups, we measured bacterial burden on day 6 for combinational experiments.
In the bacteraemia lethal model, mice began to die on day 3, hence organs were harvested from mice on day 2.
- While the authors mentioned that they collected histopathology samples in the methods section (Section 4.8), they did not include these results in the manuscript.
Reply: Sorry, it was a typo. We did not do the histology experiment. We have deleted this term.
- The language of the manuscript needs improvement.
Reply: We have improved the language of the manuscript.
Reviewer 4 Report
Comments to the author
The study is quite impressive and scientifically elaborated. I advise some minor corrections and it can be accepted after revising the following points:
- Please give in the introduction section more details about the compound M21.
- You have a high percentage of similarity rate in the material and methods part especially in the following sections:
* 4.2 Minimum inhibitory concentration (MIC) tests
*4.3 Disk diffusion and lux assays
* 4.5 Adherence assay and invasion assay
- I suggest for authors to verify the parameters and conditions of the analysis and the machine information (name, manufacturers ...).
- Please, improve the quality of figures.
- Ensure the uniformity in the units (e.g. mg/mL, µL) throughout the MS.
- Correct some English mistakes before publication. Spelling should be revised thoroughly, many spelling mistakes detected.
- Check the references in accordance with the journal style.
Author Response
The study is quite impressive and scientifically elaborated. I advise some minor corrections and it can be accepted after revising the following points:
- Please give in the introduction section more details about the compound M21.
Reply: M21 is an anti-virulence compound which we have identified previously (Gao, P.; Ho, P.L.; Yan, B.; Sze, K.H.; Davies, J.; Kao, R.Y.T. Suppression of Staphylococcus aureus virulence by a small-molecule compound. Proc Natl Acad Sci U S A 2018, 115, 8003-8008). We have added the information of M21 in the introduction part of the manuscript.
- You have a high percentage of similarity rate in the material and methods part especially in the following sections:
* 4.2 Minimum inhibitory concentration (MIC) tests
*4.3 Disk diffusion and lux assays
* 4.5 Adherence assay and invasion assay
Reply: we have rephrased these three parts.
- I suggest for authors to verify the parameters and conditions of the analysis and the machine information (name, manufacturers ...).
Reply: we have revised this information.
- Please, improve the quality of figures.
Reply: High quality figures were submitted in this revised version.
- Ensure the uniformity in the units (e.g. mg/mL, µL) throughout the MS.
Reply: We have revised it.
- Correct some English mistakes before publication. Spelling should be revised thoroughly, many spelling mistakes detected.
Reply: We have revised it.
- Check the references in accordance with the journal style.
Reply: We have checked and revised the references in accordance of the journal style.
Round 2
Reviewer 1 Report
Accept
Author Response
Nil
Reviewer 3 Report
The authors have put good effort into improving the overall presentation and addressing the issues raised, but there are some points that need clarification.
The authors have tried to address why the antibiotic-induced immune response was suppressed in the discussion section, but it is still unclear.
The primary aim of this study was to assess the impact of anti-virulence agent M21 on staphylococcal virulence factor induced by the subinhibitory concentration of the antibiotic. For such a study, a virulent strain should have been employed. On the contrary, the authors claim that the strain used in this study was not very virulent, therefore, they had to use a high dose for inducing infection. In the light of this explanation, the whole purpose of the study seems to be defeated.
The manuscript has two spellings for bacteremia (bacteraemia and bacteremia). Please make it consistent.
